COMMUNICATIONS

# Critical mingling and universal correlations in model binary active liquids

Nicolas Bain[1] & Denis Bartolo[1]

Ensembles of driven or motile bodies moving along opposite directions are generically reported to self-organize into strongly anisotropic lanes. Here, building on a minimal model of self-propelled bodies targeting opposite directions, we first evidence a critical phase transition between a mingled state and a phase-separated lane state specific to active particles. We then demonstrate that the mingled state displays algebraic structural correlations also found in driven binary mixtures. Finally, constructing a hydrodynamic theory, we single out the physical mechanisms responsible for these universal long-range correlations typical of ensembles of oppositely moving bodies.

[1] Univ Lyon, Ens de Lyon, Univ Claude Bernard, CNRS, Laboratoire de Physique, F-69342 Lyon, France. Correspondence and requests for materials should be addressed to N.B. (email: nicolas.bain@ens-lyon.fr) or to D.B. (email: denis.bartolo@ens-lyon.fr).

Should you want to mix two groups of pedestrians, or two ensembles of colloidal beads, one of the worst possible strategies would be pushing them towards each other. Both experiments and numerical simulations have demonstrated the segregation of oppositely driven Brownian particles into parallel lanes[1–5]. Even the tiniest drive results in the formation of finite slender lanes which exponentially grow with the driving strength[5]. The same qualitative phenomenology is consistently observed in pedestrian counterflows[6–10]. From our daily observation of urban traffic to laboratory experiments, the emergence of counter-propagating lanes is one of the most robust phenomena in population dynamics, and has been at the very origin of the early description of pedestrians as granular materials[11,12]. However, a description as isotropic grains is usually not sufficient to account for the dynamics of interacting motile bodies[13–15]. From motility-induced phase separation[15], to giant density fluctuations in flocks[13,16,17], to pedestrian scattering[18,19], the most significant collective phenomena in active matter stem from the interplay between their position and orientation degrees of freedom.

In this communication, we address the phase behaviour of a binary mixture of active particles targeting opposite directions. Building on a prototypical model of self-propelled bodies with repulsive interactions, we numerically evidence two non-equilibrium steady states: a lane state where the two populations maximize their flux and phase separate, and a mixed state where all motile particles mingle homogeneously. We show that these two distinct states are separated by a genuine critical phase transition. In addition, we demonstrate algebraic density correlations in the homogeneous phase, akin to that recently reported for oppositely driven Brownian particles[20]. Finally, we construct a hydrodynamic description to elucidate these long-range structural correlations, and conclude that they are universal to both active and driven ensembles of oppositely moving bodies.

## Results

**A minimal model of active binary mixtures.** We consider an ensemble of $N$ self-propelled particles characterized by their instantaneous positions $\mathbf{r}_i(t)$ and orientations $\widehat{\mathbf{p}}_i(t) = (\cos\theta_i, \sin\theta_i)$, where $i = 1, \ldots, N$ (in all that follows $\widehat{\mathbf{x}}$ stands for $\mathbf{x}/|\mathbf{x}|$). Each particle moves along its orientation vector at constant speed ($|\dot{\mathbf{r}}_i| = 1$). We separate the particle ensemble into two groups of equal size following either the direction $\Theta_i = 0$ (right movers) or $\pi$ (left movers) according to a harmonic angular potential $\mathcal{V}(\theta_i) = \frac{H}{2}(\theta_i - \Theta_i)^2$. Their equations of motion take the simple form:

$$\dot{\mathbf{r}}_i = \widehat{\mathbf{p}}_i, \tag{1}$$

$$\dot{\theta}_i = -\partial_{\theta_i}\mathcal{V}(\theta_i) + \sum_j T_{ij}. \tag{2}$$

In principle, oriented particles can interact by both forces and torques. We here focus on the impact of orientational couplings and consider that neighbouring particles interact solely through pairwise additive torques $T_{ij}$. This type of model has been successfully used to describe a number of seemingly different active systems, starting from bird flocks, fish schools and bacteria colonies to synthetic active matter made of self-propelled colloids or polymeric biofilaments[13,21–27]. We here elaborate on a minimal construction where the particles interact only by repulsive torques. In practical terms, we choose the standard form $T_{ij} = -\partial_{\theta_i}\mathcal{E}_{ij}$, where the effective angular energy simply reads $\mathcal{E}_{ij} = -B(r_{ij})\widehat{\mathbf{p}}_i \cdot \widehat{\mathbf{r}}_{ij}$. As sketched in Fig. 1a, this interaction promotes the orientation of $\widehat{\mathbf{p}}_i$ along the direction of the centre-to-centre vector $\mathbf{r}_{ij} = (\mathbf{r}_i - \mathbf{r}_j)$: as they interact particles turn their back to each other (for example, refs 24,28–30). The spatial decay

of the interactions is given by: $B(r_{ij}) = B(1 - r_{ij}/(a_i + a_j))$, where $B$ is a finite constant if $r_{ij} < (a_i + a_j)$ and 0 otherwise. In all that follows, we focus on the regime where repulsion overcomes alignment along the preferred direction ($B > 1$). The interaction ranges $a_i$ are chosen to be polydisperse to avoid the specifics of crystallization, and we make the classic choice $a = 1$ or 1.4 for one in every two particles. Before solving equations (1) and (2), two

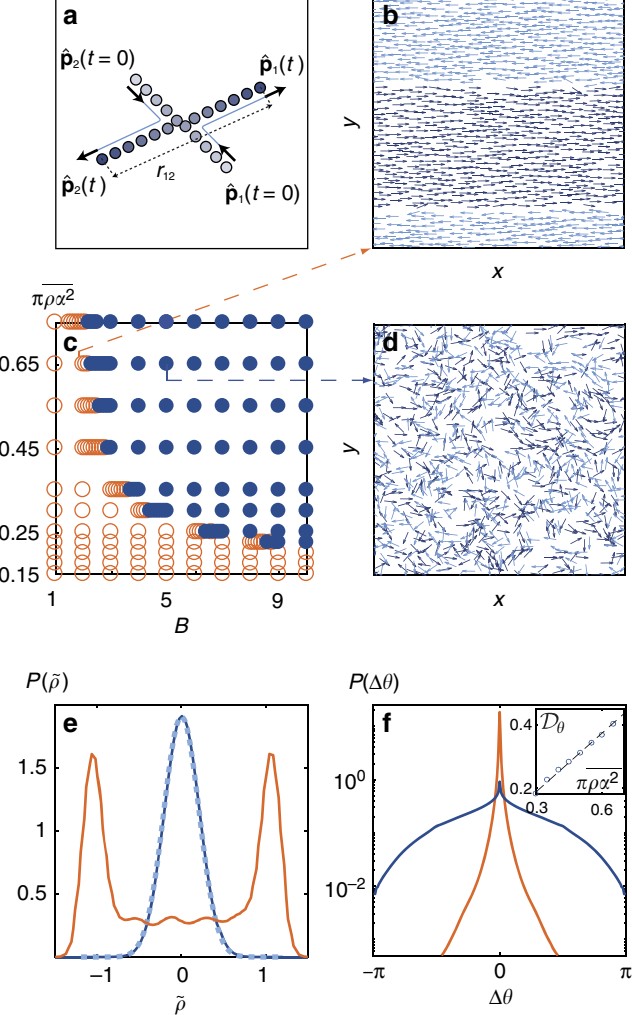

**Figure 1 | Phase behaviour.** (**a**) Trajectories of two particles interacting solely by a repulsive torque as defined in equation (2) with $B = 5$. The post-collision orientations $\widehat{\mathbf{p}}_i(t)$ are along the centre-to-centre axis $r_{ij}$. (**b,d**) Snapshots of a square window at the centre of the simulation box ($L_x = 168$, $N = 1{,}973$, $\pi\overline{\rho a^2} = 0.65$), respectively, in the lane ($B = 2$) and the homogeneous ($B = 5$) states. The arrows indicate the instantaneous position and orientation of the particles. Dark blue: right movers. Light blue: left movers. (**c**) Phase diagram. $\pi\overline{\rho a^2}$ is the particle area fraction. Filled symbols: homogeneous state. Open symbols: lanes. (**e**) Probability distribution function (p.d.f.) of the density difference $\tilde{\rho} = \rho_r - \rho_l$. Light orange line: $B = 2$, $\pi\overline{\rho a^2} = 0.65$. Dark blue line: $B = 5$, $\pi\overline{\rho a^2} = 0.65$. Dashed line: best Gaussian fit. (**f**) p.d.f. of the orientational fluctuations around the preferred orientation (lin-log plot). Same parameters and colours as in **e**. Inset: orientational diffusivity $\mathcal{D}_\theta$ in the homogeneous state at a fixed repulsion magnitude ($B = 5$) and different particle area fractions $\pi\overline{\rho a^2}$. $\mathcal{D}_\theta$ is defined as the decorrelation time of the particle orientation. In the mingled state, the velocity autocorrelation decays exponentially at short time, $\mathcal{D}_\theta$ is therefore defined without ambiguity, see also Supplementary Note 1 for a full description of the numerical computation of $\mathcal{D}_\theta$. Dashed line: best linear fit.

comments are in order. First, this model is not intended to provide a faithful description of a specific experiment. Instead, this minimal set-up is used to single out the importance of repulsion torques typical of active bodies. Any more realistic description would also include hard-core interactions. However, in the limit of dilute ensembles and long-range repulsive torques, hard-core interactions are not expected to alter any of the results presented below. Second, unlike models of driven colloids or grains interacting by repulsive forces[1,5,20], equations (1) and (2) are not invariant upon Gallilean boosts, and therefore are not suited to describe particles moving at different speeds along the same preferred direction.

**Critical mingling.** Starting from random initial conditions, we numerically solve equations (1) and (2) using forward Euler integration with a time step of $10^{-2}$, and a sweep-and-prune algorithm for neighbour summation. We use a rectangular simulation box of aspect ratio $L_x = 2L_y$ with periodic boundary conditions in both directions. We also restrain our analysis to $H = 1$, leaving two control parameters that are the repulsion strength $B$ and the overall density $\bar{\rho}$. The following results correspond to simulations with $N$ comprised between 493 and 197,300 particles.

We observe two clearly distinct stationary states illustrated in Fig. 1b,d. At low density and/or weak repulsion the system quickly phase separates. Computing the local density difference between the right and left movers $\tilde{\rho}(\mathbf{r}, t) = \rho_r(\mathbf{r}, t) - \rho_l(\mathbf{r}, t)$, we show that this dynamical state is characterized by a strongly bimodal density distribution, Fig. 1e. The left and right movers quickly self-organize into counter-propagating lanes separated by a sharp interface, Fig. 1b. In each stream, virtually no particle interact and most of the interactions occur at the interface, Supplementary Movie 1. As a result the particle orientations are very narrowly distributed around their mean value, Fig. 1f. In stark contrast, at high density and/or strong repulsion, the motile particles do not phase separate. Instead, the two populations mingle and continuously interact to form a homogeneous liquid phase with Gaussian density fluctuations, and much broader orientational fluctuations, Fig. 1d–f. This behaviour is summarized by the phase diagram in Fig. 1c.

Although phase separation is most often synonymous of first-order transition in equilibrium liquids, we now argue that the lane and the mingled states are two genuine non-equilibrium phases separated by a critical line in the $(B, \bar{\rho})$ plane. To do so, we first introduce the following orientational order parameter:

$$\langle W \rangle = \langle 1 - \cos(\theta_i - \Theta_i) \rangle_i \qquad (3)$$

$\langle W \rangle$ vanishes in the lane phase where on average all particles follow their preferred direction, and takes a non-zero value otherwise. We show in Fig. 2a how $\langle W \rangle$ increases with the repulsion strength $B$ at constant $\bar{\rho}$. For $\overline{\pi \rho a^2} = 0.65$ the order parameter averages to zero below $B_c = 2.17 \pm 0.02$, while above $B_c$ it sharply increases as $W \sim |B - B_c|^\beta$, with $\beta = 0.33 \pm 0.07$, Fig. 2b. This scaling law suggests a genuine critical behaviour. We further confirm this hypothesis in Fig. 2c, showing that the fluctuations of the order parameter diverge as $|B - B_c|^{-\gamma}$, with $\gamma = 0.64 \pm 0.07$. Deep in the homogeneous phase the fluctuations plateau to a constant value of the order of $1/N$. Finally, the criticality hypothesis is unambiguously ascertained by Fig. 2d, which shows the power-law divergence of the correlation time of $\langle W \rangle(t)$: $\tau_W \sim |B - B_c|^{-zv}$ with $zv = 1.21 \pm 0.16$.

We do not have a quantitative explanation for this critical behaviour. However, we can gain some insight from the counterintuitive two-body scattering between active particles. In the overdamped limit, the collision between two passive colloids

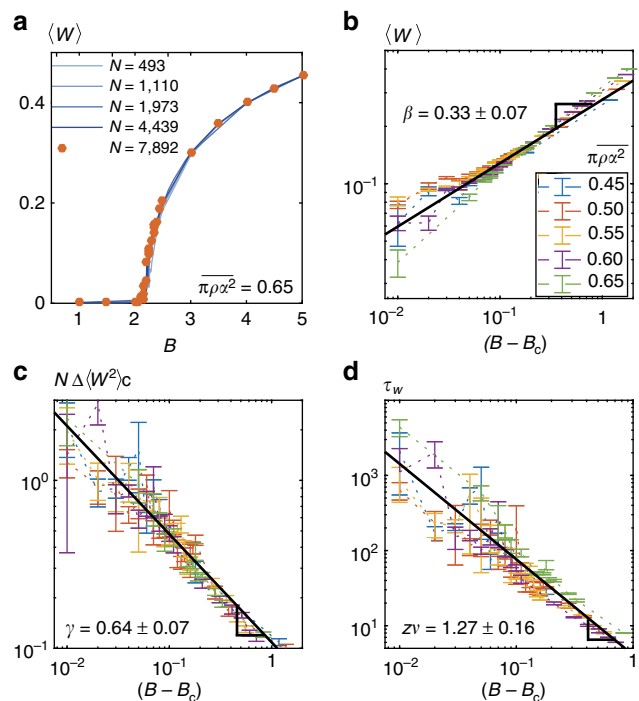

**Figure 2 | Critical transition from laned to homogeneous liquid states.** (**a,b**) Linear and log plots of the order parameter $\langle W \rangle$ defined in equation (3). (**a**): $\overline{\pi \rho a^2} = 0.65$, the bifurcation curves collapse for five system sizes. (**b–d**) Log plots at five densities for a box of length $L_x = 336$ ($N$ ranges from 5,462 to 7,892). (**c**) Fluctuations of the order parameter plotted versus $B - B_c$ for the same densities as in (**b**). The fluctuations are defined as $\Delta \langle W^2 \rangle_c \equiv \langle W^2 \rangle_c(B) - \langle W^2 \rangle_c(B \to \infty)$. (**d**) Correlation time $\tau_W$ plotted against $B - B_c$. The correlation time is defined as $\langle W(t + \tau_W) W(t) \rangle_c = \frac{1}{2} \langle W^2(t) \rangle_c$. All error bars correspond to two standard deviations. The error on the estimate of the exponents correspond to one s.d. after considering linear fits for each density.

driven by an external field would at most shift their position over an interaction diameter[31]. Here these transverse displacements are not bounded by the range of the repulsive interactions. For a finite set of impact parameters, collisions between self-propelled particles result in persistent deviations transverse to their preferred trajectories illustrated in Fig. 3 and Supplementary Note 2. This persistent scattering stems from the competition between repulsion and alignment. When these two contributions compare, bound pairs of oppositely moving particles can even form and steadily propel along the transverse direction $\hat{\mathbf{y}}$, Fig. 3b,c. We stress that this behaviour is not peculiar to this two-body setting: persistent transverse motion of bound pairs is clearly observed in simulations at the onset of laning, Supplementary Movie 2. We therefore strongly suspect the resulting enhanced mixing to be at the origin of the sharp melting of the lanes and the emergence of the mingled state.

**Long-range correlations in mingled liquids.** We now evidence long-range structural correlations in this active-liquid phase, and analytically demonstrate their universality. The overall pair correlation function of the active liquid, $g(\mathbf{r})$, is plotted in Fig. 4a. At a first glance, deep in the homogeneous phase, the few visible oscillations would suggest a simple anisotropic liquid structure. However, denoting $\alpha$ and $\beta$ the preferred direction of the populations (left or right), we find that the asymptotic behaviours of all pair correlation functions $g_{\alpha\beta}(x, y = 0)$ decay algebraically as $|1 - g_{\alpha\beta}(x, 0)| \sim x^{-v_x}$ with $v_x \sim 1.5$, Fig. 4b. This power-law

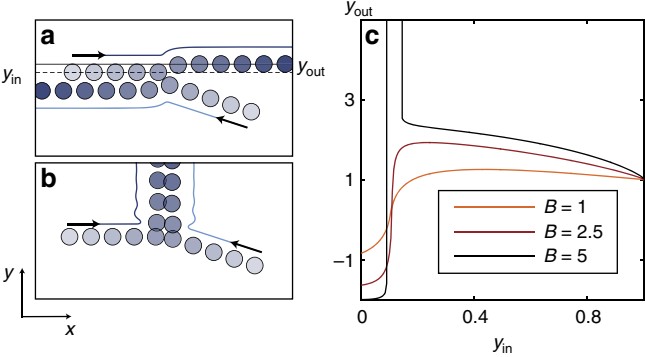

**Figure 3 | Collision between left and right movers. (a,b)** Trajectories of two colliding particles in the presence of an alignment field. The trajectories before contact are prolongations of the incoming orientations, both interactions and alignment field are only turned on at contact. **(a)** Scattering trajectory for $B = 5$, and $y_{in} = 0.75$. $y_{in}$ (resp. $y_{out}$) is the initial (resp. final) vertical position of the right mover with respect to the contact point. $y_{in}$ (resp. $y_{out}$) is represented by the dashed line (resp. plain line). **(b)** Example of collision resulting in a strong and persistent deviation along the transverse direction ($B = 5$, $y_{in} = 0.125$). **(c)** The transverse displacement $y_{out}$ is plotted as a function of the impact parameter $y_{in}$ as defined in **a**, for different values of the repulsion strength $B$. Initial conditions: a right mover with an orientation $\theta_r = 0$ and a left mover with $\theta_l = \pi - \pi/10$ are vertically placed at $+y_{in}$ and $-y_{in}$. Their $x$ coordinate is chosen so that they start interacting at $t = 0$.

behaviour is very close to that reported in numerical simulations[4] and fluctuating density functional theories of oppositely driven colloids at finite temperature[20].

**Hydrodynamic description.** To explain the robustness of these long-range correlations, we provide a hydrodynamic description of the mingled state, and compute its structural response to random fluctuations. We first observe that the orientational diffusivity of the particles increases linearly with the average density $\bar{\rho}$ in Fig. 1f inset. This behaviour indicates that binary collisions set the fluctuations of this active liquid, and hence suggests using a Boltzmann kinetic-theory framework, for example, refs 32,33 from an active-matter perspective. In the large $B$ limit, the microscopic interactions are accounted for by a simplified scattering rule anticipated from equation (2) and confirmed by the inspection of typical trajectories (Fig. 1a). Upon binary collisions the self-propelled particles align their orientation with the centre-to-centre axis regardless of their initial orientation and external drive. Assuming molecular chaos and binary collisions only, the time evolution of the one-point distribution functions $\psi_\alpha(\mathbf{r}, \theta, t)$ reads:

$$\partial_t \psi_\alpha + \nabla \cdot [\widehat{\mathbf{p}} \psi_\alpha] + \partial_\theta \left[ \partial_\theta \left( \widehat{\mathbf{p}} \cdot \widehat{\mathbf{h}}_\alpha \right) \psi_\alpha \right] = \mathcal{I}_\alpha^{coll}. \quad (4)$$

The convective term on the l.h.s stems from self-propulsion, the third term accounts for alignment with the preferred direction $\widehat{\mathbf{h}}_\alpha = \widehat{\mathbf{x}}$ (resp. $-\widehat{\mathbf{x}}$) for the right (resp. left) movers. Using the simplified scattering rule to express the so-called collision integral on the r.h.s, we can establish the dynamical equations for the density fluctuations $\delta\rho_\alpha$ around the average homogeneous state (see Methods section for technical details). Within a linear response approximation, they take the compact form:

$$\partial_t \delta\rho_\alpha(\mathbf{r}, t) + \nabla \cdot \left( \mathbf{J}_\alpha + \widetilde{\mathbf{J}} \right) = 0, \quad (5)$$

where $\mathbf{J}_\alpha$ describes the convection and the collision-induced diffusion of the $\alpha$ species, and $\widetilde{\mathbf{J}}$ is the coupling term, crucial to the

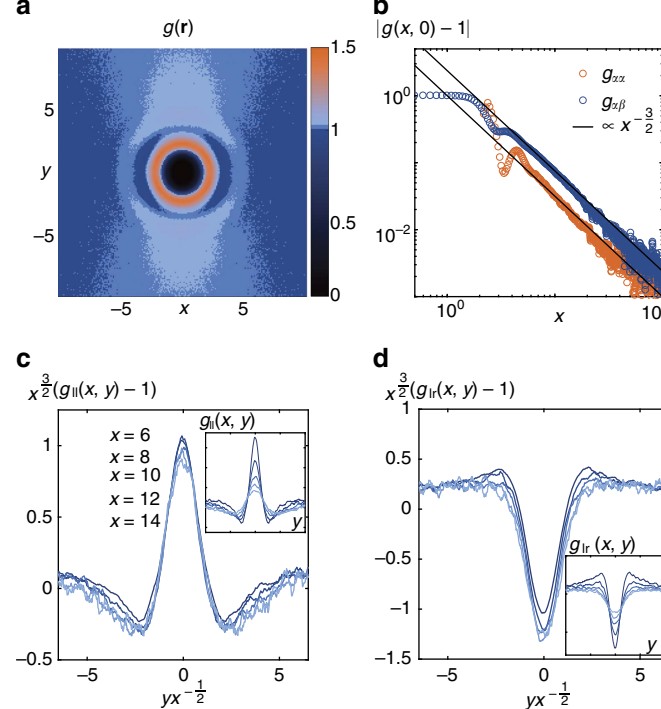

**Figure 4 | Structural correlations. (a)** Overall pair correlation function deep in the homogeneous phase ($B = 5$). **(b)** Plot of the longitudinal decay of the density auto- (light orange) and cross- (dark blue) correlation functions at $y = 0$. Black lines: algebraic decay $x^{-\frac{3}{2}}$. **(c,d)** Collapse of the pair correlations once rescaled by the universal $x^{-\frac{3}{2}}$ power law and plotted as a function of the rescaled distance $y/\sqrt{x}$. Insets: bare correlations. The good collapse of the rescaled curves supports the validity of the scaling deduced from the linearized fluctuating hydrodynamics. $B = 5$, $\overline{\pi\rho a^2} = 0.65$ and $L_y = 84$ for all panels. $N = 197,300$ particles in **b** and $N = 31,566$ in **a,c,d**.

anomalous fluctuations of the active liquid:

$$\mathbf{J}_\alpha = v_0 \widehat{\mathbf{h}}_\alpha \delta\rho_\alpha - \mathbf{D} \cdot \nabla\delta\rho_\alpha, \quad (6)$$

$$\widetilde{\mathbf{J}} = -\tilde{v}\widehat{\mathbf{h}}_\alpha \delta\bar{\rho} - \widetilde{\mathbf{D}} \cdot \nabla\delta\bar{\rho}. \quad (7)$$

The two anisotropic diffusion tensors $\mathbf{D}$ and $\widetilde{\mathbf{D}}$ are diagonal and their expression is provided in Supplementary Note 3 together with all the hydrodynamic coefficients. $\widetilde{\mathbf{J}}$ is a particle current stemming from the fluctuations of the other species and has two origins. The first term arises from the competition between alignment along the driving direction $\widehat{\mathbf{h}}_\alpha$ and orientational diffusion caused by the collisions: the higher the local density $\bar{\rho}$, the smaller the longitudinal current. The second term originates from the pressure term $\propto \nabla\bar{\rho}$: a local density gradient results in a net flow of both species (see Methods section for details). This diffusive coupling is therefore generic and enters the description of any binary compressible fluid. Two additional comments are in order. First, this prediction is not specific to the small-density regime and is expected to be robust to the microscopic details of the interactions. As a matter of fact, the above hydrodynamic description is not only valid in the limit of strong repulsion and small densities discussed above but also in the opposite limit, where the particle density is very large while the repulsion remains finite as detailed in Supplementary Note 5. Second, the robustness of this hydrodynamic description could have been anticipated using conservation laws and symmetry considerations, as done for example, in ref. 16 for active flocks.

Here the situation is simpler, momentum is not conserved and no soft mode is associated to any spontaneous symmetry breaking. As a result the only two hydrodynamic variables are the coupled (self-advected) densities of the two populations[34]. The associated mass currents are constructed from the only two vectors that can be formed in this homogeneous but anisotropic setting: $\mathbf{h}_\alpha$ and $\nabla \delta \rho_\alpha$. These simple observations are enough to set the functional form of equations (5)–(7).

By construction the above hydrodynamic description alone cannot account for any structural correlation. To go beyond this mean-field picture we classically account for fluctuations by adding a conserved noise source to equation (5) and compute the resulting density-fluctuation spectrum[13]. At the linear response level, without loss of generality, we can restrain ourselves to the case of an isotropic additive white noise of variance $2T$ (Supplementary Note 4). Going to Fourier space, and after lengthy yet straightforward algebra, we obtain in the long wavelength limit:

$$\langle |\delta\rho_\alpha(\mathbf{q})|^2 \rangle \propto \frac{q_y^4 (D_y + \tilde{D}_y)^2 + q_x^2 (v_0 - \tilde{v})^2}{q_y^4 D_y (D_y + 2\tilde{D}_y) + q_x^2 v_0 (v_0 - 2\tilde{v})} \qquad (8)$$

with $\delta\rho(\mathbf{q}) = \int \delta\rho(\mathbf{r}) \exp(-i\mathbf{q}\cdot\mathbf{r}) d\mathbf{r}$, and where $\langle \cdot \rangle$ is a noise average. The cross-correlation $\langle \delta\rho_\alpha(\mathbf{q}) \delta\rho_\beta(-\mathbf{q}) \rangle$ has a similar form, Supplementary Note 4. Even though the above hydrodynamic description qualitatively differs from that of driven colloids, they both yield the same fluctuation spectra[20]. A key observation is that the structure factor given by equation (8) is non-analytic at $q = 0$. Approaching $q = 0$ from different directions yields different limits, which is readily demonstrated noting that $\langle |\delta\rho_\alpha(q_x, q_y = 0)|^2 \rangle$ and $\langle |\delta\rho_\alpha(q_x = 0, q_y)|^2 \rangle$ are both constant functions but have different values. The non-analyticity of equation (8) in the long wavelength limit translates in an algebraic decay of the density correlations in real space. After a Fourier transform, we find: $\langle |\rho_\alpha(0,0)\rho_\alpha(x,0)| \rangle = |1 - g_{\alpha\alpha}(x)| \sim x^{-3/2}$, in agreement with our numerical simulations of both self-propelled particles, Fig. 4b, and driven colloids[4,20]. Beyond these long-range correlations it can also be shown (Supplementary Note 4) that the pair correlation functions take the form $|1 - g_{\alpha\beta}(x,y)| \sim x^{-3/2} \, \mathcal{C}(y/x^{1/2})$ again in excellent agreement with our numerical findings. Figure 4c,d indeed confirm that the pair correlations between both populations are correctly collapsed when normalized by $x^{-3/2}$ and plotted versus the rescaled distance $y/x^{1/2}$.

## Discussion

Different non-equilibrium processes can result in algebraic density correlations with different power laws, for example, ref. 35. We thus need to identify the very ingredients yielding universal $x^{-\frac{3}{2}}$ decay, or equivalently structure factors of the form $\langle |\delta\rho_\alpha(\mathbf{q})|^2 \rangle \propto (q_y^4 + a^2 q_x^2)/(q_y^4 + b^2 q_x^2)$ found both in active and driven binary mixtures. We first recall that this structure factor has been computed from hydrodynamic equations common to any system of coupled conserved fields in a homogeneous and anisotropic setting (regardless of the associated noise anisotropy,[35] and Supplementary Note 4). The structure factor is non-analytic as $q \to 0$, and the density correlations algebraic, only when $a \neq b$. Inspecting equation (8), we readily see that this condition is generically fulfilled as soon as the coupling current $\tilde{\mathbf{J}}$ is non-zero. In other words, as soon as the collisions between the particles either modify their transverse diffusion $(\tilde{\mathbf{D}} \cdot \nabla \delta\bar{\rho})$, or their longitudinal advection $(\tilde{v}\hat{\mathbf{h}}_\alpha \delta\bar{\rho})$. Both ingredients are present in our model of active particles (equation (5)) and, based on symmetry considerations, should be generic to any driven

binary mixtures with local interactions. Another simple physical explanation can be provided to account for the variations of the pair correlations in the transverse direction shown in Fig. 4c,d and also reported in simulations of driven particles[20]. Self-propulsion causes the particles to move, on average, at constant speed along the $x$-direction while frontal collisions induce their transverse diffusion. As a result the $x$-position of the particles increase linearly with time, and their transverse position increases as $\sim t^{1/2}$. We therefore expect the longitudinal and transverse correlations to be related by a homogeneous function of $y/x^{1/2}$ in steady state as observed in simulations of both active and driven particles. Altogether these observations confirm the universality of the long-range structural correlations found in both classes of non-equilibrium mixtures.

In conclusion, we have demonstrated that the interplay between orientational and translational degrees of freedom, inherent to motile bodies, can result in a critical transition between a phase separated and a mingled state in binary active mixtures. In addition, we have singled out the very mechanisms responsible for long-range structural correlations in any ensemble of particles driven towards opposite directions, should they be passive colloids or self-propelled agents.

## Methods

**Boltzmann kinetic theory.** Let us summarize the main steps of the kinetic theory employed to establish equations (5)–(7). The so-called collision integral on the r.h.s of equation (4) includes two contributions, which translate the behaviour illustrated in Fig. 1a:

$$\mathcal{I}_\alpha^{\text{coll}} = \mathcal{D}_{\text{in}} \rho_\alpha(\mathbf{r}) \bar{\rho}(\mathbf{r} - 2a\hat{\mathbf{p}}) - \mathcal{D}_{\text{out}} \bar{\rho}(\mathbf{r}) \psi_\alpha(\mathbf{r}, \theta). \qquad (9)$$

The first term indicates that a collision with any particle located at $(\mathbf{r} - 2a\hat{\mathbf{p}})$ reorients the $\alpha$ particles along $\hat{\mathbf{p}}(\theta)$ at a rate $\mathcal{D}_{\text{in}}$. The second term accounts for the random reorientation, at a rate $\mathcal{D}_{\text{out}}$, of a particle aligned with $\hat{\mathbf{p}}(\theta)$ upon collision with any other particle. Within a two-fluid picture, the velocity and nematic texture of the $\alpha$ particles are given by $\mathbf{v}_\alpha = \rho_\alpha^{-1} \langle \hat{\mathbf{p}} \rangle_\theta$ and $\mathbf{Q}_\alpha = \rho_\alpha^{-1} \langle \hat{\mathbf{p}}\hat{\mathbf{p}} - \frac{1}{2}\mathbb{I} \rangle_\theta$. The mass conservation relation, $\partial_t \rho_\alpha + \nabla \cdot (\rho_\alpha V_\alpha) = 0$, is obtained by integrating equation (4) with respect to $\theta$ and constrains $(2\pi\mathcal{D}_{\text{in}}) = \mathcal{D}_{\text{out}} \equiv \mathcal{D}$. The time evolution of the velocity field is also readily obtained from equation (4):

$$\partial_t(\rho_\alpha \mathbf{v}_\alpha) + \nabla \cdot \left[ \rho_\alpha \left( \frac{\mathbb{I}}{2} + \mathbf{Q}_\alpha \right) \right] = \boldsymbol{\mathcal{F}}_\alpha, \qquad (10)$$

where the second term on the l.h.s is a convective term stemming from self-propulsion. The force field $\boldsymbol{\mathcal{F}}_\alpha$ on the r.h.s. of equation (10) reads: $\boldsymbol{\mathcal{F}}_\alpha = \rho_\alpha (\frac{1}{2} - \mathbf{Q}_\alpha) \cdot \hat{\mathbf{h}}_\alpha - (a\mathcal{D}\rho_\alpha)\nabla\bar{\rho} - (\mathcal{D}\bar{\rho})\rho_\alpha \mathbf{v}_\alpha$. The first term originates from the alignment of particles along the $\hat{\mathbf{h}}_\alpha$ direction, the second term is a repulsion-induced pressure, and the third one echoes the collision-induced rotational diffusivity of the particles. An additional closure relation between $\mathbf{Q}_\alpha$, $\mathbf{v}_\alpha$ and $\rho_\alpha$ is required to yield a self-consistent hydrodynamic description. Deep in the homogeneous phase, we make a wrapped Gaussian approximation for the orientational fluctuations in each population[24,36]. This hypothesis is equivalent to setting $\mathbf{Q}_\alpha = |\mathbf{v}_\alpha|^4 (\hat{\mathbf{v}}_\alpha \hat{\mathbf{v}}_\alpha - \frac{1}{2}\mathbb{I})$ (refs 24,37). As momentum is not conserved, the velocity field is not a hydrodynamic variable; in the long wavelength limit the velocity modes relax much faster than the (conserved) density modes. We therefore ignore the temporal variations in equation (10) and use this simplified equation to eliminate $\mathbf{v}_\alpha$ in the mass conservation relation, leading to the mass conservation equation (5).

**Data availability.** The data that support the findings of this study are available from the corresponding author upon request.

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

## Acknowledgements

We acknowledge support from ANR grant MiTra and Institut Universitaire de France (D.B.). We acknowledge valuable comments and suggestions by V. Démery and H. Löwen.

## Author contributions

D.B. designed the research. N.B. performed the numerical simulations. D.B. and N.B. performed the theory, discussed the results and wrote the paper.

## Additional information

**Competing interests:** The authors declare no competing financial interests.

**Publisher's note**: 

