## [Peer Review File · Nature Communications]

Reviewers' comments:

Reviewer #1 (Remarks to the Author):

This paper describes a new nonequilibrium critical transition from a mixed to an unmixed state as a function of the strength of the interparticle coupling. Using computer simulations of a simple driven model system, the authors identify a continuous transition and extract some effective critical exponents. Also long-ranged structural correlations with an inverse exponent $3/2$ are obtained and a hydrodynamic description is put forward which reproduces this behavior. In principle, this is an interesting paper but there are some points which should be addressed before a further consideration and final recommendation of the paper can be done.

In detail:

1) The model misses the translational collisions completely since there is no repulsive core in the interactions. The speed of the particles/pedestrians is constant along their dynamics (Equation (1)). Therefore some of the important parts of the physics is missing in this model. In particular the Peclet number is assumed to be constant in the model resp. it can be trivially scaled out and is therefore fixed.

2) Figures 1b and 1d show two snapshots below and above the transition. A mixed laned state is shown below the transition which is stable at low densities and small couplings B. Unfortunately the ideal gas limit is not shown in Figure 1c. There are two routes to achieve this limit. either the density goes to zero or the coupling B vanishes. In this limit, the interaction is virtually zero. Why is there an unmixed state at all as the oppositely driven particles do not feel each other? Is this an artifact of the periodic boundary conditions? Or in other words: why do weakly interacting particles demix?

3) Figure 4 b: The statistical uncertainties in the data limit the validity of the $x^{-3/2}$ decay to x values smaller than a decade (e.g. half a decade for g_{α}). Better data are needed to make a firm conclusion here.

4) How was the orientational diffusion coefficient D_{θ} (shown in Figure 1f and discussed on page 3) obtained numerically? And how does the density dependence compare with polymer theory (for example Doi's tube model for hard rods)?

Reviewer #2 (Remarks to the Author):

In this article, the authors explore a model of lane formation in binary mixtures of self-driven particles. They show, fairly convincingly, that the model exhibits critical behavior as a function of the particle density and the interaction strength. They measure the critical exponents and the behavior of the correlation function. A hydrodynamic description is also constructed.

This is a fairly interesting paper, and is technically sound (as far as I can tell). But the model being studied is fairly artificial, and the authors do little to acknowledge the limitations of their conclusions. Indeed, the model being studied is kind of a strange one. Particles interact through a "mutual torque" that tries to align them back-to-back, and they are kept on track by an angular harmonic potential. While the authors claim that "this type of interaction is not only common ... but relevant to synthetic active particles", I find little evidence to support this claim, and I think this sentence is misleading. The cited references [19, 21, 22] examine models that are very different from the one considered here (and are different from each other). The strangeness of the model is perhaps highlighted by the fact that increasing the interaction strength *_destroys_* lane formation rather than promoting it.

The authors' results are largely presented as universal (for example, in the abstract), presumably in the sense of being independent of the microscopics of the model. But clearly not all of them are (for example, the phase diagram is certainly specific to this model). Are the critical exponents universal in some sense? The reader is left with the large question as to which of the results of this paper can be generalized beyond the fairly arbitrary and specific microscopic details. Relatively little attention is paid to this question.

So I suggest that the authors tone down their implications of universality and address more directly the question of which of their results are relatively independent of microscopic details. For example, the title should probably become something like "Critical mingling and universal correlations in a model of binary active liquids", and similar edits should be made to abstract to emphasize that they are analyzing a specific model.

Most pressing, it seems like reference [16] makes specific universal predictions for the critical exponents. Are they valid here? If not, then why not? The authors briefly mention that "the same hydrodynamic theory ... could have been anticipated using conservation law and symmetry" and cite reference [16]. But does this mean that their model is literally described by the results of reference [16] in the long-wavelength limit? This is a big and important question. I think a revised manuscript really needs to address this question, as well as the larger question of universality of their results, in a concerted way. The comparison with reference [16] seems crucial.

So, in summary, I recommend that the paper be re-considered for publication after extensive revision.

A few more specific technical comments and questions:

1. In the caption of figure 1, part (c) is incorrectly referred to as "(b)".
2. The orientational diffusivity $D_{\{\theta\}}$ seems to show no critical behavior or discontinuity through the transition. Is there a simple reason to understand why this is the case?
3. Fig. 2a shows the order parameter as a function of B for different system sizes L . While the transition being shown is very sharp, at finite system size the vanishing of the order parameter at some $B = B_c$ should be smeared a bit. This smearing near $B = B_c$ should enable one to do a critical scaling analysis to determine the localization length $\xi \sim B^{-\nu}$. In particular, it should be possible to write the curve $W(B, L)$ as a universal function of L/ξ , or equivalently of $(B - B_c)L^{1/\nu}$.

In other words, by plotting W as a function of $(B - B_c)L^{1/\nu}$ for different values of ν , it

should be possible to extract the correlation length exponent ν by finding the value which causes all curves to collapse onto each other.

Is such an analysis possible? If so, it should be added, since this gives an independent estimate of ν . If not, the authors should comment on why it is not possible.

4. The exponent γ is defined as the negative of its usual value: $(\text{fluctuations}) \sim |B - B_c|^{-\gamma}$. (In the comment below I use the usual definition.)

5. It seems that there should be scaling relations between the different critical exponents. In particular, there is the standard relation $\nu d = 2\beta + \gamma$. Using the authors' results, this relation would seem to give $\nu = 0.64$, from which one can determine the dynamical exponent $z = 1.89$.

Is this correct? If so, it should be added, and presumably compared to the result of reference [16], which (seemingly) predicts $z = 6/5$.

6. I think the comparison to reference [5] on page 3 is misleading. Reference [5] considered a very different model (driven, oppositely-charged particles with a Yukawa interaction). There would seem to be no *a priori* reason to expect it to be comparable to the present results. But I suppose this goes back to my central critique.

7. The authors make a brief comment in the discussion about the necessary ingredients to give the $1/x^{3/2}$ behavior in the correlation function. But this comment is only 1 or 2 sentences, which are dense and hard to interpret. I strongly suggest that the authors elaborate these sentences into a more careful, thorough, and friendly discussion. The question of the universality of the paper's results is, in my opinion, central to its appeal. So any definitive comments the authors can make should be emphasized and explained clearly.

Reviewer #3 (Remarks to the Author):

I read this article by Bain and Bartolo with interest. The formation of lanes in active and driven systems is a problem that has attracted regular attention in recent years.

The work presented here is a detailed study of one particular numerical model of laning, which descends (as far as I can tell) from a Vicsek model with repulsion. As such, one feature that distinguishes it from other approaches is that the particle speed is always constant, regardless of orientation.

Numerically, the authors show that laning occurs only *below* a critical repulsion strength, which decreases with increasing density. The authors show that this transition has features consistent with a second order phase transition, and is quite different from the analogous transition in more conventional driven brownian particles (Glanz and Loewen, 2012 ref. [5] - note that I am *not* one of the authors). The authors pair the numerics with a sophisticated hydrodynamic coarse-graining of the system in the disordered (non-laning) phase, and show long-range density correlations consistent with the numerics, and also results for active brownian particles.

This is solid work, and in particular the hydrodynamic theory is very sophisticated. However, I do not think that Nature Communications is the right journal for this paper:

- The numerical model is very unusual, in that particle collisions only affect the orientation, and there is no direct repulsion mechanism. In other words, from overdamped Langevin dynamics one would expect an additional term $\sum_j F_{ij}$ in equation 1, and it's not there. This makes this model not appropriate for active colloids, cells, droplets and similar particles that are expected to follow overdamped Langevin dynamics. It is still appropriate for intelligent agents, like

pedestrians, birds, or maybe robots.

- Can the authors give an example of a real physical system where equations 1 would be a reasonable approximation of the dynamics?

- The numerical model has no noise, either rotational (D_r) or translational (D), and there is no indication on the how robust the results are if noise were added. The authors also never vary the self-propulsion speed away from $v_0=1$. Ref. [5] indicated that the Peclet number $Pe \sim v_0 / D_r$ plays a crucial role in the laning transition of brownian particles. There is no way to put the results obtained here into context and compare them.

- In a way, it is natural for the laning to disappear when the scattering cross section B is increased. B is not a repulsion in a traditional sense, as other effects in addition to the scattering would appear and interfere with the central mechanism that destroys the laning phase, particle pairs taking off at an angle. With physical repulsion, these pairs heading into oncoming traffic, so to speak, would quickly be pushed back into their lanes.

- The hydrodynamic theory is very nice, and very sophisticated. However, are the long-range density correlations in the disordered state really that interesting? The theory is not able to say anything about the laning transition, or the properties of the stable lane state.

In my opinion, this is a valuable contribution to the laning literature, but there are lingering questions about the model, and the generality of the results.

This should be published in a more specialised journal. In its current presentation, a lot of material has also been abbreviated to the point of being hard to follow, and even the supplementary material is quite dense. These issues would be solved by rewriting this as a longer paper.

Response to reviewers

REVIEWER 1

We thank the referee for her/his encouraging report which helped us improving both the presentation of the model, and the physical discussion of our main results. We have addressed her/his main comments in our revised version.

1. The model misses the translational collisions completely since there is no repulsive core in the interactions. The speed of the particles/pedestrians is constant along their dynamics (Equation (1)). Therefore some of the important parts of the physics is missing in this model. In particular the Peclet number is assumed to be constant in the model resp. it can be trivially scaled out and is therefore fixed.

It was obvious from the three referee reports that a clearer introduction of the model was required. In line with these comments, we have extensively modified the presentation of the model and clarified its underlying framework. (The main changes and additions are presented in blue).

We now clearly explain that we intentionally build on a minimal model to explore the impact of orientational interactions on the dynamics of binary active mixtures. We do not intend to provide an accurate description of a specific experimental system. We agree that this task would most certainly require including translational collisions via e.g hard-core repulsion. However, the impact of repulsion forces has already been extensively studied not only in the context of driven colloids but also in active matter (see. e.g the extensive body of work on the so called Motility Induced Phase Separation introduced by Cates and Tailleur). We have therefore decided to single out the impact of orientational interactions on the phase behavior of active binary mixtures.

In order to better contextualize this class of model we provide references to three review articles accessible to a broad audience and systematically discuss the physical meaning of this description (Marchetti et al, Rev. Mod. Phys. 2013, by Vicsek and Zafeiris, Physics Reports 2012, Cavagna and Giardina, Annual Review of Condensed Matter Physics 2014). We stress however that we do not merely repeat simulations previously introduced in active-matter physics. We here focus (i) on a minimal model where repulsion torques prevail over e. g. alignment interactions or hard-core repulsion (until now this situation has been overlooked by physicists) and (ii) on binary populations with opposite orientations (a situation that has never been considered to our knowledge).

The same response applies to the effect of the Peclet number. To begin with we have decided to focus on a microscopic model at "0 temperature" associated with a rich a phenomenology. The extension of this research to finite Peclet numbers is definitely a very interesting question which ranks high in our to do list. We hope that this manuscript will stimulate research along those lines by other groups.

2. Figures 1b and 1d show two snapshots below and above the transition. A mixed laned state is shown below the transition which is stable at low densities and small couplings B . Unfortunately the ideal gas limit is not shown in Figure 1c. There are two routes to achieve this limit. either the density goes to zero or the coupling B vanishes. In this limit, the interaction is virtually zero. Why is there an unmixed state at all as the oppositely driven particles do not feel each other? Is this an artifact of the periodic boundary conditions? Or in other words: why do weakly interacting particles demix?

We thank the referee for pointing out this issue. The labels of the x -axis in Figure 1c were indeed misleading. In the corrected version we now clearly show that the x -axis of the phase diagram starts at $B = 1$, and not $B = 0$ where the particles would not interact with each other. We have also further stressed in the main text that we only focus on the "strong" interaction regime where repulsion overcomes the torque aligning the particles along their preferred direction ($B > 1$).

The only ideal-gas limit considered in this article therefore corresponds to $\bar{\rho} \rightarrow 0$. Repelling particles would clearly phase separate in this limit. Take for instance the extreme situation where two particles target opposite directions in a box (periodic or not). After a couple of collisions, repulsion would shift the transverse positions of the particles at a distance at least equal to the interaction range. In this absorbing state the two motile particles would endlessly move along straight lines thereby forming the simplest possible form of a lane "state". It is therefore expected that a minimal packing fraction is required to destabilize the lanes as confirmed by the phase diagram in Fig. 1c (here $\sim 30\%$).

3. Figure 4 b: The statistical uncertainties in the data limit the validity of the $x^{-3/2}$ decay to x values smaller than a decade (e.g. half a decade for $g_{\alpha\alpha}$). Better data are needed to make a firm conclusion here.

We have followed the referee's suggestion and conducted simulations in boxes ten times longer, keeping the width constant. The resulting gain in the statistics makes the agreement with the $x^{-3/2}$ prediction clearer.

In addition the agreement between our numerical findings and analytic theory is not only supported by Fig. 4b but also by a prediction of the large-distance anisotropy of the pair correlation function. To better support our conclusions we have moved the corresponding discussion from a Supplementary Note to the main text (see also new figures 4c and 4d). Our theory predicts that the pair correlations functions are homogeneous functions of the form $|1 - g_{\alpha\beta}(x, y)| \sim x^{-3/2}\mathcal{C}(y/x^{1/2})$. This functional form is shown to very nicely rescale our numerical data and therefore further confirms the relevance of our model. A simple physical argument is provided to explain this behavior in the discussion section.

4. How was the orientational diffusion coefficient \mathcal{D}_θ (shown in Figure 1f and discussed on page 3) obtained numerically? And how does the density dependence compare with polymer theory (for example Doi's tube model for hard rods)?

Rotational diffusion is defined as the inverse of the decorrelation time of the particle orientation. We now provide more details about the measurement of \mathcal{D}_θ in a supplementary Note.

The linear dependence of \mathcal{D}_θ with the particle density can actually be understood using an argument very similar to that used to explain the diffusivity of a dilute Lorentz gaz. Upon each collision the particle orientation experiences uncorrelated kicks. Doubling the particle density, the number of collisions per unit time doubles, therefore yielding angular diffusion with a diffusivity increasing linearly with the particle density. The Doi model for crowded hard rods actually predicts an opposite trend. The reason is that, in this different context, hard-core interactions hinder *thermal* rotational diffusion.

REVIEWER 2

We thank the referee for her/his thorough review of our manuscript. We have taken her/his comments into consideration to considerably clarify the main text and supplementary document. We hope she/he will find this revised version suitable for publication. A detailed answer to her/his main comments and seven specific points is provided below.

General comments

This is a fairly interesting paper, and is technically sound (as far as I can tell). But the model being studied is fairly artificial, and the authors do little to acknowledge the limitations of their conclusions. Indeed, the model being studied is kind of a strange one. Particles interact through a "mutual torque" that tries to align them back-to-back, and they are kept on track by an angular harmonic potential. While the authors claim that "this type of interaction is not only common ... but relevant to synthetic active particles ", I find little evidence to support this claim, and I think this sentence is misleading. The cited references [19, 21, 22] examine models that are very different from the one considered here (and are different from each other). The strangeness of the model is perhaps highlighted by the fact that increasing the interaction strength destroys lane formation rather than promoting it.

It was obvious from the three referee reports that a clearer introduction of the model was required. In line with these comments, we have extensively modified the presentation of the model and clarified its underlying framework (The main changes and additions are presented in blue).

However, we respectfully disagree with the referee about the singularity of our model. Pointwise polar particles moving at constant speed and coupled by interaction torques is a very well established paradigm in active-matter physics. Since the introduction of seminal Vicsek model 20 years ago, this type of description has been successfully employed to account for the gross features of a host of seemingly different physical and biological systems ranging from wave propagation in bird groups [22] and bacterial swarms [25] to flocking transitions in ensembles of self-propelled colloids [23].

Nonetheless, we do appreciate that Nature Communications is not only intended to active-matter physicists. For this reason, taking into consideration the referees' comments, we both provide references to three review articles accessible to a broad audience and systematically discuss the physical meaning of this theoretical description in this revised version. We stress however that we do not merely repeat a simulation previously introduced in active-matter physics. We here focus (i) on a minimal model where repulsion torques prevail over e. g. alignment interactions or hard-core repulsion (until now this situation has been mostly overlooked by physicists) and (ii) on binary population with opposite orientations. A situation that has never been considered to our knowledge.

The authors' results are largely presented as universal (for example, in the abstract), (...) The reader is left with the large question as to which of the results of this paper can be generalized beyond the fairly arbitrary and specific microscopic details. Relatively little attention is paid to this question. So I suggest that the authors tone down their implications of universality and address more directly the question of which of their results are relatively independent of microscopic details. For example, the title should probably become something like "Critical mingling and universal correlations in a model of binary active liquids", and similar edits should be made to abstract to emphasize that they are analyzing a specific model.

We have taken this point very seriously. Starting from the title and the abstract, we set out to distinguish between the phenomenology specific to our prototypical model and results universal to a number of active and driven systems. Our main universal prediction concerns the algebraic structural correlations in the homogeneous (mingled) state. They do *not* rely on self-propulsion nor on coupling to orientational degrees of freedom. The same type of correlations have been numerically reported in driven ensemble of passive colloids. Our hydrodynamic theory explains the very ingredients required to observe this universal behaviour which does not stem from criticality.

Most pressingly, it seems like reference [16] makes specific universal predictions for the critical exponents. Are they valid here? If not, then why not? The authors briefly mention that "the same hydrodynamic theory ... could have been anticipated using conservation law and symmetry" and cite reference [16]. But does this mean that their model is literally described by the results of reference [16] in the long-wavelength limit? This is a big and important question. A think a revised manuscript really needs to address this question, as well as the larger question of universality of their results, in a concerted way. The comparison with reference [16] seems crucial.

We thank the referee for pointing this possible ambiguity. We have clarified and extended the paragraph where we

refer to the seminal paper by Toner and Tu to avoid any possible confusion (Below Eqs. 6 and 7). We refer to this article in order to provide an example of a hydrodynamic theory of active matter constructed solely on the basis of symmetry arguments and conservation laws. We did not mean to imply that the Toner and Tu theory applies to our system.

Unlike flocking models where a conserved density field couples to a slow orientational soft mode, the present system involves two coupled conserved fields. Even in the long wave-length limit the two models are intrinsically different and correspond to two different classes of active materials. The exponents discussed by Toner and Tu have therefore no reason to be the same as the critical exponents measured at the mingling transition (note also that the Toner and Tu exponents do not apply to the flocking transition which is (in most cases) first order, instead they describe the scale-free fluctuations of a broken-symmetry active fluid).

Specific points

1. In the caption of figure 1, part (c) is incorrectly referred to as "(b)".

This has been corrected in the revised manuscript.

2. The orientational diffusivity \mathcal{D}_θ seems to show no critical behavior or discontinuity through the transition. Is there a simple reason to understand why this is the case?

This is a good point. This remark helped us clarifying Fig. 1f. The value of \mathcal{D}_θ is plotted only in the mingled regime. In the lane state \mathcal{D}_θ is very heterogeneous. It takes a finite value at the interfaces and virtually vanishes in the bulk. We have also added a section in the Supplementary document where we further detail the measurement of this quantity.

3. Fig. 2a shows the order parameter as a function of B for different system sizes L . While the transition being shown is very sharp, at finite system size the vanishing of the order parameter at some $B = B_c$ should be smeared a bit. This smearing near $B = B_c$ should enable one to do a critical scaling analysis to determine the localization length $\xi \propto B^{-\nu}$. In particular, it should be possible to write the curve $W(B, L)$ as a universal function of L/ξ , or equivalently of $(B - B_c)L^{1/\nu}$. In other words, by plotting W as a function of $(B - B_c)L^{1/\nu}$ for different values of ν , it should be possible to extract the correlation length exponent ν by finding the value which causes all curves to collapse onto each other. Is such an analysis possible? If so, it should be added, since this gives an independent estimate of ν . If not, the authors should comment on why it is not possible.

We feel that this one is more than a specific point! We have actually tried to perform a finite-size scaling analysis. However, we do not have sufficient precision close to the critical point to make any predictive measurement. We do believe that a comprehensive finite-size analysis would deserve a separate paper on its own. In addition to be notoriously demanding the analysis is here complexified by the intrinsic system anisotropy. A full analysis would require establishing a scaling function $W(B, L_x, L_y)$ to infer two exponents ν_x and ν_y thereby making the analysis even more demanding in terms of computation time. As our most robust (universal) prediction concerns the mingled state, away from the transition, we do not feel that establishing the values of all the exponents and possible hyper-scaling relations are central to our discussion, and would like to leave it to further studies.

4. The exponent γ is defined as the negative of its usual value: (fluctuations) $\approx |B - B_c|^{-\gamma}$. (In the comment below I use the usual definition.)

We now use the standard notation suggested by the referee.

5. It seems that there should be scaling relations between the different critical exponents. In particular, there is the standard relation $\nu d = 2\beta + \gamma$. Using the authors' results, this relation would seem to give $\nu = 0.64$, from which one can determine the dynamical exponent $z = 1.89$. Is this correct? If so, it should be added, and presumably compared to the result of reference [16], which (seemingly) predicts $z = 6/5$.

We have now clarified the difference with the scaling relations established by Toner and Tu for flocks. See also our answer to 3. regarding the hyperscaling relation.

6. I think the comparison to reference [5] on page 3 is misleading. Reference [5] considered a very different model (driven, oppositely-charged particles with a Yukawa interaction). There would seem to be no a priori reason to expect it to be comparable to the present results. But I suppose this goes back to my central critique.

We agree with referee 2 and referee 3. At retrospect, the comparison with reference [5] (old numbering) was indeed misleading and did not add any value to the discussion (We still credit the work of Glanz and Löwen in the introduction as this article establishes a solid result for driven systems).

7. The authors make a brief comment in the discussion about the necessary ingredients to give the $1/x^{3/2}$ behavior in the correlation function. But this comment is only 1 or 2 sentences, which are dense and hard to interpret. I strongly suggest that the authors elaborate these sentences into a more careful, thorough, and friendly discussion. The question of the universality of the paper's results is, in my opinion, central to its appeal. So any definitive comments the authors can make should be emphasized and explained clearly.

Following the referee's advice we have extended this discussion central to our paper. We have also elaborated on the universality of the hydrodynamic description itself putting more emphasis on physics than on the technical aspects of the kinetic theories detailed in a supplementary note (pp 4 and 5). In order to further emphasize the robustness of our predictions we have also included additional results about the functional form of the pair correlation functions and provided simple physical arguments to quantitatively account for their anisotropy. Part of these results were presented in the first version of the Supplementary Notes.

REVIEWER 3

We thank the reviewer for her/his positive comments about our manuscript. We answer all her/his questions below and have improved the manuscript in line with all her/his remarks. Taking into account all the suggestions made by the three referees we resubmit a significantly revised version of the original text.

1. The numerical model is very unusual, in that particle collisions only affect the orientation, and there is no direct repulsion mechanism. In other words, from overdamped Langevin dynamics one would expect an additional term $\sum_j F_{ij}$ in equation 1, and it's not there. This makes this model not appropriate for active colloids, cells, droplets and similar particles that are expected to follow overdamped Langevin dynamics. It is still appropriate for intelligent agents, like pedestrians, birds, or maybe robots.

It was obvious from the three referee reports that a clearer introduction of the model was required. In line with these comments, we have extensively modified the presentation of the model and clarified its underlying framework (The main changes and additions are presented in blue).

Let us address more specifically the main concern of Reviewer 3. This type of description was indeed originally introduced to account for the dynamics of animals groups (starting e.g. from the Vicsek model). However it was already a simplification, intelligent entities are also subject to interaction forces as well. Yet these forces are unnecessary to account for a host of collective phenomena, such as wave propagation and long range correlations in bird groups see e.g. the reviews by Cavagna and Giadina Annual Review of Condensed Matter Physics (2014). The same observation holds for synthetic systems. Several large scale phenomena arising in ensembles of synthetic and bacterial active fluids have been successfully described using the same type of orientational dynamics, assuming constant speed and considering only orientational interactions (see e.g. refs [24, 25,26] in the revised manuscript).

Here, we do not intend to provide an accurate description of a specific experimental system which would most certainly require including translational collisions. In addition, the impact of repulsion forces has already been extensively studied not only in the context of driven colloids but also in active matter (see. e.g the extensive body of work on the so called Motility Induced Phase Separation introduced by Cates and Tailleur). Our primary goal is here to single out the impact of orientational interactions on the rich yet overlooked phase behavior of active binary mixtures.

2. Can the authors give an example of a real physical system where equations 1 would be a reasonable approximation of the dynamics?

In principle any active system where the mean propulsion speed is large compared to the speed modifications induced by collisions could be modelled by similar equations, possibly with additional torques. Let us be even more specific, as confirmed by preliminary experiments in our group, colloidal rollers [23] made of paramagnetic beads could be a simple realization of a motile system where the interactions are dominated by repulsion at constant speed. In addition we stress that the model is expected to apply to dilute populations of intelligent creatures which have been proven to change their orientation to avoid collisions (as opposed to translating keeping their orientation fixed).

3. The numerical model has no noise, either rotational D_r , or translational D , and there is no indication on the how robust the results are if noise were added. The authors also never vary the self-propulsion speed away from $v_0=1$. Ref. [5] indicated that the Peclet number $Pe \approx v_0/D_r$ plays a crucial role in the laning transition of brownian particles. There is no way to put the results obtained here into context and compare them.

We do agree with referee 3 (and referee 2). At retrospect, the comparison with reference [5] was indeed misleading and did not add any value to the discussion (We still credit the work of Glanz and Löwen in the introduction as this article establishes a solid result for driven systems at finite temperature).

However we respectfully disagree with the referee on one point. Even at zero temperature this model tells us a lot about the dynamics of oppositely moving active particles, and it seems very reasonable to first investigate the limit of vanishingly small Pe number. In this asymptotic case, we are already left with a bi-dimensional phase diagram with rich features. We believe that investigating the impact of intrinsic translational and rotational diffusion is an interesting subject which we would like to leave for future research. Note however that the general form of the hydrodynamic description extensively discussed in the last sections will not be modified at finite temperature. Therefore, we do expect the universal long-range correlations of the mingled state to be insensitive to changes in Pe .

4. In a way, it is natural for the laning to disappear when the scattering cross section B is increased. B is not a repulsion in a traditional sense, as other effects in addition to the scattering would appear and interfere with the

central mechanism that destroys the laning phase, particle pairs taking off at an angle. With physical repulsion, these pairs heading into oncoming traffic, so to speak, would quickly be pushed back into their lanes.

We do agree with this analysis which we actually put forward in the main text. However, see above, B also quantifies a *physical* repulsion even though it derives from a torque acting on a polar body rather than a force.

5. The hydrodynamic theory is very nice, and very sophisticated. However, are the long-range density correlations in the disordered state really that interesting? The theory is not able to say anything about the laning transition, or the properties of the stable lane state.

We thank the referee for his encouraging comments about our theoretical model. We believe that the fluctuations in the mingled state are an interesting phenomena. As further stressed in this revised version, these long-range structural correlations are universal to a number of non-equilibrium binary mixtures, from motile particles to driven colloids. In contrast the criticality of the transition might depend on the specifics of the collisions as anticipated by the referee in his previous comment. We therefore put a stronger emphasis on the former phenomenology. It is however true that the present theory does not capture correctly the transition, as it has been explicitly derived deep in the mingled phase.

6. In my opinion, this is a valuable contribution to the laning literature, but there are lingering questions about the model, and the generality of the results.

We have taken the comments and suggestions made by the referees very seriously into consideration and set out to further clarify and expand our manuscript. We believe we have addressed all the valuable questions and concerns raised by the referee in her/his report and hope she/he will find this revised version suitable for publication.

REVIEWERS' COMMENTS:

Reviewer #1 (Remarks to the Author):

The authors have done a great deal to improve their manuscript according to the comments of the referees. This is after all indeed an interesting paper. However, after having received the reply of the authors I realized that the model studied is indeed a bit special. I have therefore still doubts whether the criteria required for Nature Communications in terms of broad relevance for a general community are met but I leave this decision to the editor.

Reviewer #2 (Remarks to the Author):

In this resubmission the authors have done a thorough and conscientious job of addressing the concerns raised by the referees. I think the paper can be accepted and published as-is.

Reviewer #3 (Remarks to the Author):

The authors have carefully answered all the questions of the three referees, and substantially revised their manuscript. I do not have any more concerns except for the central one:

Is this a reasonable or even relevant model for any natural system? All three referees have pointed out the unnatural features of the model, in particular the absence of translational collision coupling.

The authors in their revised manuscript now state

"Firstly, this model is not intended to provide a faithful description of a specific experiment. Instead, this minimal setup is used to single out the importance of repulsion torques typical of active bodies."

and in the response to my comments

"[...] it seems very reasonable to first investigate the limit of vanishingly small Pe number"

I respectfully disagree, currently it is impossible to tell if any of the features of the model presented here are experimentally relevant. The exception is are indeed the correlation in the mingling phase, but, again, is that all that interesting?

My previous conclusion hasn't changed: This is very solid work, but it belongs in a more specialised journal than Nature Communications.